# Association of suicidal behavior with exposure to suicide and suicide attempt: A systematic review and multilevel meta-analysis

Nicole T. M. Hill[1,2]*, Jo Robinson[1], Jane Pirkis[3], Karl Andriessen[3], Karolina Krysinska[1], Amber Payne[1,4], Alexandra Boland[1], Alison Clarke[1], Allison Milner[5†], Katrina Witt[1], Stephan Krohn[6,7‡], Amit Lampit[6,7,8‡]*

1 Orygen, Centre for Youth Mental Health, University of Melbourne, Parkville, Victoria, Australia, 2 Telethon Kids Institute, Perth, Western Australia, Australia, 3 Centre for Mental Health, Melbourne School of Population and Global Health, University of Melbourne, Parkville, Victoria, Australia, 4 Northeastern University, Boston, Massachusetts, United States of America, 5 Centre for Health Equity, School of Population and Global Health, University of Melbourne, Parkville, Victoria, Australia, 6 Department of Neurology, Charité–Universitätsmedizin Berlin, Berlin, Germany, 7 Berlin School of Mind and Brain, Humboldt-Universität zu Berlin, Berlin, Germany, 8 Department of Psychiatry, University of Melbourne, Parkville, Victoria, Australia

† Deceased.
‡ These authors are joint senior authors on this work.
* nicole.hill@telethonkids.org.au (NTMH); amit.lampit@unimelb.edu.au (AL)

**Data Availability Statement:** All relevant data are within the manuscript and its Supporting

## Abstract

### Background

Exposure to suicidal behavior may be associated with increased risk of suicide, suicide attempt, and suicidal ideation and is a significant public health problem. However, evidence to date has not reliably distinguished between exposure to suicide versus suicide attempt, nor whether the risk differs across suicide-related outcomes, which have markedly different public health implications. Our aim therefore was to quantitatively assess the independent risk associated with exposure to suicide and suicide attempt on suicide, suicide attempt, and suicidal ideation outcomes and to identify moderators of this risk using multilevel meta-analysis.

### Methods and findings

We systematically searched MEDLINE, Embase, PsycINFO, CINAHL, ASSIA, Sociological Abstracts, IBSS, and Social Services Abstracts from inception to 19 November 2019. Eligible studies included comparative data on prior exposure to suicide, suicide attempt, or suicidal behavior (composite measure—suicide or suicide attempt) and the outcomes of suicide, suicide attempt, and suicidal ideation in relatives, friends, and acquaintances. Dichotomous events or odds ratios (ORs) of suicide, suicide attempt, and suicidal ideation were analyzed using multilevel meta-analyses to accommodate the non-independence of effect sizes. We assessed study quality using the National Heart, Lung, and Blood Institute quality assessment tool for observational studies. Thirty-four independent studies that presented 71 effect sizes (exposure to suicide: $k = 42$, from 22 independent studies; exposure

Information files. Summary data used in the analyses are provided as S1 Data.

**Funding:** NH is a PhD student and was supported by the Australian Rotary Health PhD Partnership Scholarship (https://australianrotaryhealth.org.au). JR was supported by a National Health and Medical Research Council (NHMRC) Career Development Fellowship (APP1142348, https://www.nhmrc.gov. au). KA was supported by a NHMRC Early Career Fellowship (APP1157796, https://www.nhmrc.gov. au). AM was supported by a Victorian Health and Medical Research Fellowship, Department of Health and Human Services (https://www.vic.gov. au). KW was supported by a post-doctoral fellowship awarded by the American Foundation for Suicide Prevention (PDF-0-145-16, https://afsp. org). SK was supported by the German Federal Ministry for Education and Research (BMBF grant 13GW0206D, https://www.bmbf.de). AL was supported by a NHMRC-Australian Research Council Dementia Research Development Fellowship (APP1108520, https://www.nhmrc.gov. au). The funders had no role in study design, data collection and analysis, decision to publish, or preparation of the manuscript.

**Competing interests:** The authors have declared that no competing interests exist.

**Abbreviations:** OR, odds ratio.

to suicide attempt: $k = 19$, from 13 independent studies; exposure to suicidal behavior (composite): $k = 10$, from 5 independent studies) encompassing 13,923,029 individuals were eligible. Exposure to suicide was associated with increased odds of suicide (11 studies, $N = 13,464,582$; OR = 3.23, 95% CI = 2.32 to 4.51, $P < 0.001$) and suicide attempt (10 studies, $N = 121,836$; OR = 2.91, 95% CI = 2.01 to 4.23, $P < 0.001$). However, no evidence of an association was observed for suicidal ideation outcomes (2 studies, $N = 43,354$; OR = 1.85, 95% CI = 0.97 to 3.51, $P = 0.06$). Exposure to suicide attempt was associated with increased odds of suicide attempt (10 studies, $N = 341,793$; OR = 3.53, 95% CI = 2.63 to 4.73, $P < 0.001$), but not suicide death (3 studies, $N = 723$; OR = 1.64, 95% CI = 0.90 to 2.98, $P = 0.11$). By contrast, exposure to suicidal behavior (composite) was associated with increased odds of suicide (4 studies, $N = 1,479$; OR = 3.83, 95% CI = 2.38 to 6.17, $P < 0.001$) but not suicide attempt (1 study, $N = 666$; OR = 1.10, 95% CI = 0.69 to 1.76, $P = 0.90$), a finding that was inconsistent with the separate analyses of exposure to suicide and suicide attempt. Key limitations of this study include fair study quality and the possibility of unmeasured confounders influencing the findings. The review has been prospectively registered with PROSPERO (CRD42018104629).

## Conclusions

The findings of this systematic review and meta-analysis indicate that prior exposure to suicide and prior exposure to suicide attempt in the general population are associated with increased odds of subsequent suicidal behavior, but these exposures do not incur uniform risk across the full range of suicide-related outcomes. Therefore, future studies should refrain from combining these exposures into single composite measures of exposure to suicidal behavior. Finally, future studies should consider designing interventions that target suicide-related outcomes in those exposed to suicide and that include efforts to mitigate the adverse effects of exposure to suicide attempt on subsequent suicide attempt outcomes.

## Author summary

### Why was this study done?

- Exposure to suicidal behavior in others has been linked to increased risk of suicidal behavior, but it is not known whether the association differs between types of exposure (suicide versus suicide attempt) or different outcome measures of suicidal thoughts and behaviors.

- Distinguishing the relationships of different exposure types with outcomes is important for the development of targeted interventions and public health approaches to suicide prevention.

### What did the researchers do and find?

- We conducted a systematic review and meta-analysis of 34 studies that investigated the independent associations between exposure to different types of suicidal behavior and subsequent suicide, suicide attempt, and suicidal ideation outcomes.

- We showed that exposure to suicide is associated with increased odds of both suicide and suicide attempt, but found limited evidence of an association with suicidal ideation. Exposure to suicide attempt was associated with increased odds of suicide attempt only.

- For exposure to suicide, degree of relationship (i.e., whether the suicide exposure occurred in a relative as compared to a friend or acquaintance) did not materially affect the magnitude of the association. The odds of suicidal behavior (i.e., including attempted suicide) were, however, greater when the exposure occurred in a relative.

### What do these findings mean?

- Exposure to suicide is associated with greater odds of suicide and suicide attempt. Yet, exposure to suicide attempt is associated with increased odds of suicide attempt only.

- Researchers and public health practitioners should refrain from combining suicide, suicide attempt, and suicidal ideation into composite measures of suicide exposures and outcomes.

- We recommend that future public health policy include the potential adverse effects of exposure to suicide attempt.

## Introduction

Suicide attempt and suicide are leading causes of global morbidity and mortality. Approximately 800,000 people die by suicide annually [1], of which about one-third are under the age of 30 [2]. The prevalence of suicide attempt is significantly greater than that of suicide death and is associated with heightened risk of later death by suicide [3,4] as well as psychosocial adversities that persist later in life [5]. For every suicide death, it is estimated that approximately 135 people are affected [6]. Over the course of a lifetime, the proportion of people exposed to the suicide of a relative, friend, or acquaintance is approximately 21% [7]. Exposure to suicide has been linked to increased risk of physical disease and adverse mental health including depression, posttraumatic stress disorder, and complicated grief [8,9]. The deleterious effects associated with exposure to suicide may also render some people, particularly adolescents and young adults, at increased risk of suicide and suicide attempt [10].

Combined, the large number of people exposed to suicide and the potential increased risk of suicide-related outcomes (suicide, suicide attempt, and suicidal ideation) in others mean that exposure to suicide is a significant public health concern [1]. This is reflected in several national suicide prevention strategies that recommend postvention interventions for those bereaved by suicide [11], as well as several international frameworks for the prevention of suicide-related contagion, and the management of suicide and self-harm clusters [12–15]. These public health strategies have largely focused on exposure to suicide, despite a growing body of evidence that suggests that exposure to suicide attempt, the behavior most proximal to suicide, may also be associated with increased risk of suicide-related outcomes [16–19].

Distinguishing between the potential independent effects of exposure to suicide and suicide attempt is important since measures of morbidity and mortality have markedly different public health implications. Yet evidence regarding the independent effects of exposure to suicide and suicide attempt on subsequent suicide-related outcomes is unclear. A systematic review

and meta-analyses by Geulayov and colleagues [20] showed that exposure to suicide and exposure to suicide attempt of a parent were associated with increased risk of suicide and suicide attempt in offspring. However, the authors pooled mean effect sizes across subgroups within studies and did not take into account the dependencies between effect sizes, an approach that may distort the results of the meta-analyses [21]. Another systematic review by Crepeau-Hobson and Leech [19] reported that both exposure to suicide and exposure to suicide attempt were associated with subsequent suicide-related behavior among friends or acquaintances. But the authors did not adequately control for studies that reported lifetime prevalence, leaving the causal direction between exposure to suicide attempt and subsequent suicide-related outcomes unclear.

Lack of guiding evidence has impeded translation of the evidence into practice. For example, it is not currently clear which populations may be at risk, nor whether the risk differs across outcomes involving suicide, suicide attempt, and suicidal ideation. Sveen and Walby [22] found inconclusive evidence supporting a relationship between exposure to suicide and increased risk of suicide-related behavior in others. However, the authors combined studies reporting exposure in relatives and friends or acquaintances, which may incur different suicide risk. More recently, systematic reviews that investigated exposure to suicide in friends and acquaintances have reported a positive association between exposure to suicide and subsequent suicide-related outcomes [19,23]. Yet, as noted previously, the causal direction between exposure and outcome measures were confounded by the inclusion of studies that reported lifetime prevalence of exposure and outcome measures. Lastly, some studies included outcome measures that combined suicidal ideation with suicide attempt [24,25] or combined exposure to suicide and exposure to suicide attempt as a composite measure of exposure to suicidal behavior [26–28]. Composite measures of exposure to suicidal behavior prevent us from identifying whether the observed effect is influenced by a true association or the result of a cumulative effect.

Consequently, the effects of prior exposure to suicide and suicide attempt on suicide-related outcomes have not been reliably quantified, and the factors that moderate this risk are not currently known. We therefore aimed to conduct a systematic review and multilevel meta-analysis investigating the independent association between prior exposure to suicide, suicide attempt, and suicidal behavior (composite measure—suicide or suicide attempt) and subsequent suicide, suicide attempt, and suicidal ideation in relatives, friends, and acquaintances. In doing so, we aimed to quantify the association between exposure to suicide and suicide attempt and the full range of suicide-related outcomes, and to identify whether factors such as relationship to the person who engaged in the initial suicidal act, age of the study population, and study design characteristics moderate this risk. By using multilevel meta-analyses, we were able to account for dependencies among multiple effect sizes taken from the same cohort within a study, an extremely common and challenging aspect of conducting meta-analyses of epidemiological studies [29].

## Methods

This work adheres to PRISMA (Preferred Reporting Items for Systematic Reviews and Meta-Analyses) [30] and MOOSE (Meta-analysis of Observational Studies in Epidemiology) [31] guidelines (S1 Text) and was prospectively registered with PROSPERO (CRD42018104629). Deviations from the protocol include the use of exposure to suicidal behavior (composite) and statistical analyses using multilevel meta-analyses. The association between exposure to suicide and suicide attempt and grief and mental health outcomes will be reported in a separate systematic review and meta-analysis.

## Electronic search strategy

We searched MEDLINE, Embase, PsycINFO, Cumulative Index to Nursing and Allied Health Literature (CINAHL), Applied Social Sciences Index and Abstracts (ASSIA), Sociological Abstracts, International Bibliography of the Social Sciences (IBSS), and Social Services Abstracts from inception through 19 November 2019 for observational studies examining the effects of exposure to suicide, suicide attempt, or suicidal behavior on 1 or more outcomes relating to suicide, suicide attempt, or suicidal ideation. Search terms relating to exposure to suicide and suicide attempt as well suicide bereavement, suicide contagion, and suicide clusters were combined using Boolean logic (S2 Text). The search was not limited by time, location, year of publication, or language (articles written in a language other than English were translated using Google Translate). Additional articles were identified by scanning the reference lists of included articles and previous reviews. One author (NTMH) conducted the initial search and screening of titles and abstracts. Three authors independently screened the full text of each potentially eligible article (NTMH, AB, KA, and KW). Discrepancies were resolved by the first author (NTMH), who also contacted the corresponding authors of primary studies for additional information.

## Study selection and eligibility criteria

Eligible studies reported dichotomous events (both the exposure and outcome were reported as having occurred or not occurred, yielding a 2 × 2 matrix) or odds ratios (ORs) for exposure to suicide, suicide attempt, or suicidal behavior and subsequent suicide, suicide attempt, or suicidal ideation. Exposure to suicide, suicide attempt, or suicidal behavior was determined from self-reported measures, informant interviews, official records (such as hospital admission records), or data linkage to death certificates. Outcomes involving suicide, suicide attempt, or suicidal ideation were determined from self-reported measures, informant interviews, or official records, such as death certificates, coroner reports, or hospital admission records. Cohort, case–control, and cross-sectional study designs were eligible if the study was reported in a peer-reviewed journal and the temporal sequence between the exposure and outcome was specified. For cross-sectional studies, the temporal sequence between exposure and outcome was established if the outcome measurement occurred after the exposure (e.g., the study asked participants if they had made a suicide attempt after exposure to the suicide of another). Participants of any age who were exposed to prior suicide or suicide attempt were eligible if the sample was mainly, or solely, drawn from the general population, as opposed to a clinical or other high-risk population (e.g., inpatients or prison detainees). Eligible control groups included individuals who did not report prior exposure to suicide, suicide attempt, or suicidal behavior in others.

Studies were excluded if findings from a non-exposed (control) group were not reported, or the control group was composed of participants exposed to other modes of death (e.g., accident or natural causes). Studies that reported estimates of lifetime prevalence as well as studies that did not establish the temporal sequence between exposure and suicide-related outcomes (e.g., the study reported 12-month prevalence of the outcome, but prior exposure to suicide was not indicated) were excluded. Finally, studies that reported outcomes following exposure to media reports of suicide (including fictional and non-fictional portrayals) or non-suicidal self-injury were excluded.

## Data collection and coding

Two independent reviewers (NTMH and KK) extracted data using a standardized data collection form. A description of the a priori moderators of risk included in the study are presented

in S1 Table. Dichotomous data were favored over ORs. When dichotomous events were not available, unadjusted ORs were recorded. For studies with multiple follow-up time points, only data from the longest time point were extracted [32]. Studies that included participants from the same population during overlapping time periods (e.g., nationwide data registry studies that reported suicide deaths from overlapping time periods) were included only if the studies reported different relationships (e.g., relatives and friends and acquaintances) or different suicide-related outcomes. When studies combined measures of exposure to the suicide of a relative or friend, we contacted primary authors for disaggregated data. If these data were not available, the relationship between the exposed individual and the individual(s) who engaged in suicidal behavior was determined by a majority rule (the relationship that occurred most frequently as indicated in >50% of the total sample). Similarly, if the age of participants included a combination of youths and adults, the age of the population was categorized in favor of the age group that exceeded 50% of the overall population. Study-level data are provided as S1 Data.

## Multilevel meta-analysis rationale and data analysis

Since 16/34 (47%) studies reported multiple exposure and/or outcome measures in the same sample of participants, the assumption of independent estimates for a traditional meta-analysis was not met. We therefore used a 3-level meta-analysis, which parallels traditional random effects meta-analyses. The main difference is that dependent effect sizes (due to multiple subgroups or outcome measures within studies) are nested within studies (level 2) before these are pooled across studies (level 3). Thus, $\tau^2_{(2)}$ is the variance within studies while $\tau^2_{(3)}$ is the variance between studies. This approach allows for the investigation of heterogeneity not only between but also within studies [33]. For clarity, we use the general term "multilevel" throughout to describe our analyses.

We conducted a multilevel meta-analysis with the maximum likelihood estimation method using the metaSEM package [34] for R version 3.6.0. For the main analysis, we used dichotomous event data to calculate the pooled OR with the accompanying 95% confidence interval (CI) for risk of suicide, suicide attempt, and suicidal ideation within exposed and non-exposed individuals. When event data were not available, we used unadjusted ORs. Meta-analyses were conducted separately for exposure to suicide, suicide attempt, and suicidal behavior. Heterogeneity was quantified as variance in true effects within ($\tau^2_{(2)}$) and between ($\tau^2_{(3)}$) studies. We also report the $I^2$ statistic, which represents the proportion of variance in true effects out of total variance for each level (i.e., $I^2_{(2)}$ and $I^2_{(3)}$), along with its 95% confidence interval. Maximum likelihood mixed-effects analyses were used to examine effect moderators via subgroup analysis and to explain heterogeneity (quantified as $R^2$) for each level. Since the multilevel model does not provide study-level effect estimates, forest plots present the mean OR of each study but report the pooled 3-level estimate. Small study effect ("publication bias") was assessed by visually inspecting funnel plots of mean log ORs against standard error for asymmetry [35]. When at least 10 studies were available for analysis, we formally assessed funnel plot asymmetry using a multilevel analogue of Egger's test of the intercepts [36].

## Risk of bias and quality assessment

Study quality was assessed using the National Heart, Lung, and Blood Institute quality assessment tool for observational studies [37]. The original tool contains 14 criteria that determine potential sources of bias in the study population and selection of participants, outcome and exposure measurement, blinding, confounding, and attrition. An overall rating of "good,"

"fair," or "poor" was provided for each independent study. Three independent reviewers conducted assessments (NTMH, AP, and AC), and any discrepancies were settled through discussion and finalized by the primary author (NTMH).

## Results

### Study selection

The initial search identified 21,868 records, of which 8,320 were duplicates. A total of 13,548 records were screened based on title and abstract (Fig 1). The full-text versions of 760 records were assessed, 10 of which were obtained from searching the reference lists of existing reviews. The authors of 6 studies were contacted [38–43], and information or additional data provided for 2 studies [39,41]. A total of 167 records reported outcomes relating to suicide, suicide attempt, or suicidal ideation. Of these, 73 articles reported lifetime prevalence estimates, 35 studies involved overlapping populations or superseded time points, and 2 studies did not report ORs or accompanying effect sizes: These articles were therefore excluded from the meta-analysis. One study [44] was excluded because it reported an OR of 36.4, and 1 study [45] was excluded because it reported an OR of 18; both studies were prone to artifacts introduced by quasi-separation (S2 Table; S3 Text). The final dataset included 34 independent studies, which comprised 71 effect sizes (exposure to suicide: $k = 42$ across $n = 22$ studies; exposure to suicide attempt: $k = 19$ across $n = 13$ studies; exposure to suicidal behavior: $k = 10$ across $n = 5$ studies).

### Characteristics of studies

Thirty-four studies were included in the meta-analysis ($N = 13,923,029$; Table 1). In terms of exposure to suicide, 22 studies ($N = 13,607,708$) provided a total of 42 effect sizes for suicide ($k = 24$), suicide attempt ($k = 15$), and suicidal ideation ($k = 3$). For exposure to suicide attempt, 13 studies ($N = 342,516$) provided a total of 19 effect sizes for suicide ($k = 3$) and suicide attempt ($k = 16$). For exposure to suicidal behavior (composite measure—suicide or suicide attempt), 5 studies ($N = 2,145$) provided a total of 10 effect sizes for suicide ($k = 7$) and suicide attempt ($k = 3$). Studies were from a range of geographic settings including Australia/New Zealand [46–48], North America [16,18,28,49–57], Europe [17,41,58–63], East Asia [26,27,64–69], the Middle East [39,42], and South America [70]. Overall, 20/34 studies involved youths aged 25 years or less. Overall exposure was determined by informant interviews in 14/34 (41%) studies, self-report measures in 12/34 (35%) studies, and official death records in 8/34 (24%) studies. A total of 6/34 (18%) studies reported separate effect sizes for exposure to suicide and exposure to suicide attempt, and 5/34 (15%) studies reported effect sizes for both exposure in relatives and exposure in friends. In terms of outcome measurements, most studies (23/34, 68%) used official hospital admission or death records, followed by self-report measures (10/34, 29%) and informant interviews (1/34, 3%). One study (1/34, 3%) reported outcomes for both suicide attempt and suicidal ideation following exposure to suicide. No studies reported suicidal ideation outcomes following exposure to suicide attempt or suicidal behavior. Lastly, 3 studies included exposure and outcome measurements of deliberate self-harm, irrespective of intent [48,59,62]. The remaining studies did not define suicide attempt [16,18,42,52,54,57,59,70], or defined suicide attempt as an act involving explicit intent to die [16,17,28,39,47,49–51,53,56,64].

### Study quality

Studies were most commonly rated fair (13/34) and good (13/34), followed by poor (8/34; S3 Table). The 13 good-quality studies tended to comprise cohort or case–control study designs

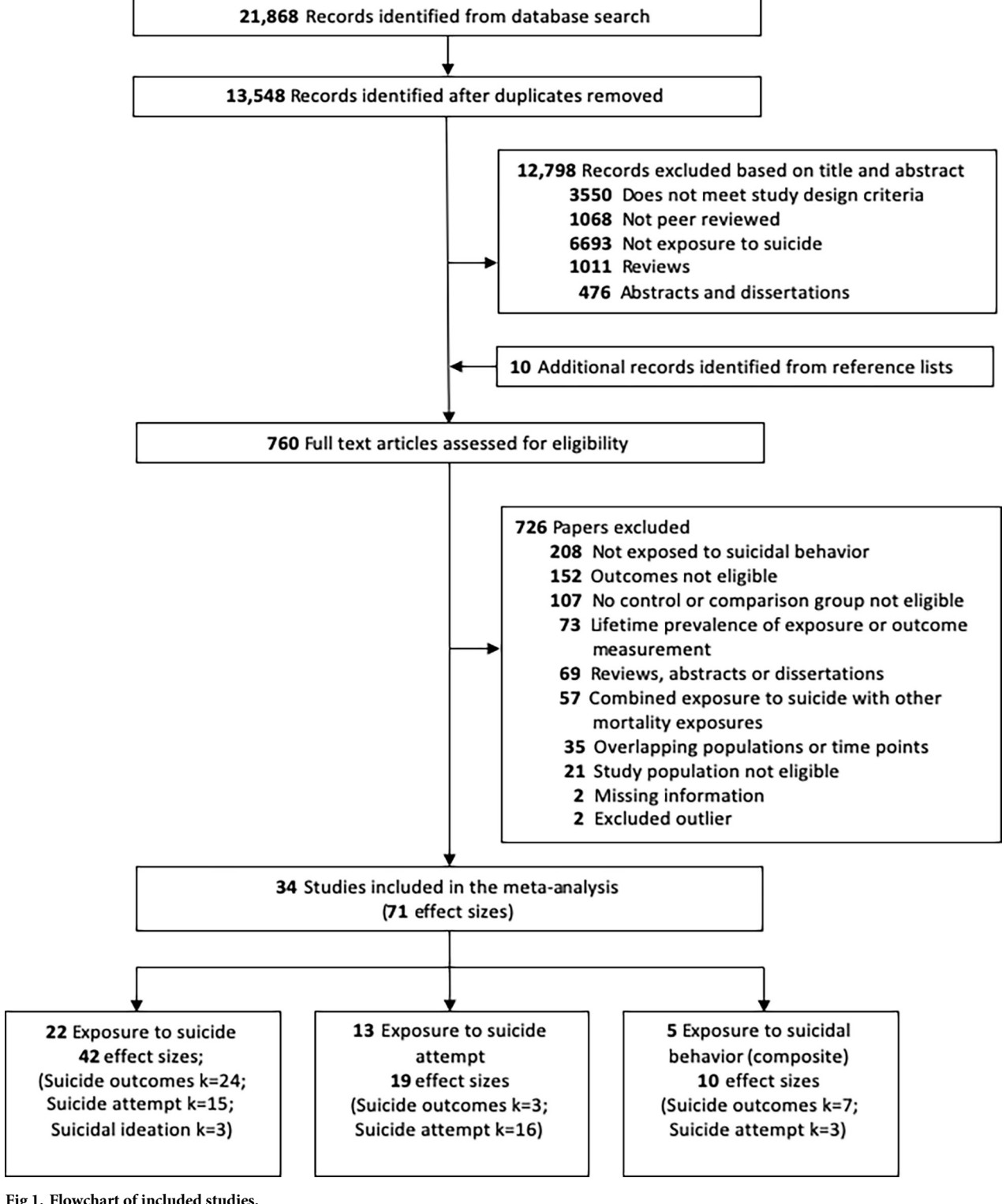

**Fig 1. Flowchart of included studies.**

and had clearly defined and valid exposure and outcome measures that were verified using official hospital or death records. The 8 studies that were rated poor tended to combine

**Table 1. Characteristics of included studies.**

| Study, location, study design | Exposed population, mean age or age range (years), percent female, total sample size | Exposure | Definition of exposure | Outcome(s) | Definition of outcome(s) | Exposure ascertainment | Study quality |
|---|---|---|---|---|---|---|---|
| Agerbo 2003 [58], Denmark, case–control | Adult[b], age range = 9–44, 24.52% female, N = 4,444,297 | Suicide of relative (any relative)[a] | Official records: Cause of death register. Exposure(s) determined by ICD codes for suicide and self-inflicted injury (ICD-8/9: E950–E959), intentional self-harm (ICD-10: X60–X84), and sequelae of intentional self-harm (ICD-10: Y870). | Suicide | Official death records: Cause of death register. Outcome(s) determined by ICD codes for suicide and self-inflicted injury (ICD-8/9: E950–E959), intentional self-harm (ICD-10: X60–X84), and sequelae of intentional self-harm (ICD-10: Y870). | The outcome occurred after the date of the exposure determined through data linkage. | Good |
| Almeida 2012 [46], Australia, cross-sectional | Adult, mean age = 70.5, age range = 60–101, 58.7% female, N = 21,290 | Suicide of relative (first-degree relative)[b] | Self-report: Participants were asked if any immediate family member had died by suicide. | Suicidal ideation | Self-report: Participants completed the Depressive Symptom Inventory Suicidality Subscale. | Determined by current suicidal ideation (persistent over the last 2 weeks). Exposure occurred at least 2 months prior. | Good |
| Brent 1996a [51], US, cohort | Youth, mean age = 20.8, 46.6% female, N = 341 | Suicide of friend or acquaintance | Informant: Suicide death in the family. | Suicide attempt | Self-report: Participants were asked if they have engaged in deliberate self-harm with intent to die. | New onset of suicide attempt since exposure. | Good |
| Brent 1996b [50], US, cohort | Youth, mean age = 20.2, 50% female, N = 44 | Suicide of relative (sibling) | Self-report: Suicide death in the family. | Suicide attempt | Self-report: Participants were asked if they have engaged in deliberate self-harm with intent to die. | New onset of suicide attempt since exposure. | Good |
| Christiansen 2011 [59], Denmark, case–control | Youth, mean age = 17.49, 78.75% female, N = 69,649 | Suicide of relative (parent) | Official records: Exposure(s) determined by ICD codes for suicide and self-inflicted injury (ICD-8/9: E950–E959) and intentional self-harm (ICD-10: X60–X84). | Suicide attempt | Hospital admission records: Outcome(s) determined by ICD codes for suicide and self-inflicted injury (ICD-8/9: E950–E959), intentional self-harm (ICD-10: X60–X84), self-poisoning event of undetermined intent (ICD-10: Y10–Y34), injury of muscle and tendon at neck level (ICD-10: S617–S619), sequelae of poisoning by drugs, medicaments and biological substances (ICD-10: T36–T60), and toxic effect of unspecified substance (ICD-10: T65). | The outcome occurred after the date of the exposure determined through data linkage. | Good |
| Gravseth 2010 [61], Norway, cohort | Adult[b], age range = 19–37, 48.82% female, N = 610,359 | Suicide of relative (parent) | Official records: Exposure(s) determined by ICD codes for suicide and self-inflicted injury (ICD-8/9: E950–E959) and intentional self-harm ICD-10: (X60–X84). | Suicide | Official death records: Outcome(s) determined by ICD codes for suicide and self-inflicted injury (ICD-9: E950–E959) and intentional self-harm (ICD-10: X60–X84). | The outcome occurred after the date of the exposure determined through data linkage. | Good |
| Giupponi 2018 [41], Italy, case–control[a] | Adult, mean age = 48.25, 38.16% female, N = 262 | Suicide of relative (any relative) | Informant: Participants were asked if there was a history of suicide in the family. Informed by at least 2 people including relatives or close friends. | Suicide | Official death records: Cause of death hospital forensic post-mortem records. | Psychological autopsy—suicide occurred after exposure. | Fair |
| Lee 2018 [66], Taiwan, cohort | Youth, 63.4% aged <17, 47.75% female, N = 438,330 | Suicide of relative (parent) | Official records: Taiwan death registry. Exposure(s) determined by ICD codes for suicide and self-inflicted injury (ICD-8/9: E950–E959), intentional self-harm (ICD-10: X60–X84), and sequelae of intentional self-harm (ICD-10: Y870). | Suicide | Official death records: Taiwan death registry. Outcome(s) determined by ICD codes for suicide and self-inflicted injury (ICD-8/9: E950–E959), intentional self-harm (ICD-10: X60–X84), and sequelae of intentional self-harm (ICD-10: Y870). | The outcome occurred after the date of the exposure determined through data linkage. | Good |

(*Continued*)

**Table 1.** (Continued)

| Study, location, study design | Exposed population, mean age or age range (years), percent female, total sample size | Exposure | Definition of exposure | Outcome(s) | Definition of outcome(s) | Exposure ascertainment | Study quality |
|---|---|---|---|---|---|---|---|
| Liu 2019 [67], China, case–control[a] | Adult, mean age = 60.87, 43.15% female, N = 380 | Suicide of relative (any relative) | Informant: Informants were asked if there was a history of suicide in the family. Informed by at least 1 relative or close friend. | Suicide | Official death records: Center for Disease Control and Prevention records of suicide. | Psychological autopsy—suicide occurred after exposure. | Fair |
| Conner 2007 [64], China, case–control | Adult[b], age range <18 to 55+ (64% aged <35), 76% female, N = 554 | Suicide of friend or acquaintance[c] | Self-report: Participants were asked if there was a history of suicide in an associate or relative. | Suicide attempt | Hospital admission records: Hospital admission for intentional suicide attempt. | All participants were hospitalized for suicide attempt at the time that prior exposure was measured. | Fair |
| Foster 1999 [60], Ireland, case–control[a] | Adult[b], age range <20 to 79 (32% aged <29), 28.2% female, N = 230 | Suicide of relative (any relative) | Informant: Informants were asked if there was a family history of suicide. Informants not indicated but were "bereaved" by suicide. | Suicide | Official death records: Coroner-determined suicide death. | Psychological autopsy—suicide occurred after exposure. | Fair |
| Gray 2014 [55], US, case–control[a] | Adult, mean age = 39.9, 32.5% female, N = 423 | Suicide of relative (any relative) | Informant: Informants were asked if there was a family history of suicide. Informed by next of kin. | Suicide | Official death records: Cause of death register, Utah Office of the Medical Examiner. | Psychological autopsy—suicide occurred after exposure. | Fair |
| Katibeh 2018 [42], Iran, case–control | Youth, mean age = 15.5, age range ≤ 18, percent female not reported, N = 300 | Suicide of relative (parent) | Self-report: Participants were asked if there was a history of suicide in their parents. | Suicide attempt | Hospital admission records: Hospital admission records for suicide attempt. | All participants were hospitalized for suicide attempt at the time that prior exposure was measured. | Poor |
| Swanson & Colman 2013 [57], Canada, cohort (cross-sectional analyses) | Youth, age range = 12–15, 50.1% female, N = 22,064 | Suicide of friend or acquaintance | Self-report: Participants were asked whether anyone in their school had died by suicide (schoolmate's suicide) and whether they personally knew anyone who had died by suicide. | Suicide attempt and suicidal ideation | Self-report (suicide attempt): Participants were asked to report the number of suicide attempts they had made in the past year, and participants were asked if they had seriously considered attempting suicide in the past year. | Prior exposure measured at baseline, and subsequent suicide attempt was based on participants who reported having made a suicide attempt within the 2-year follow-up period. | Fair |
| Tidemalm 2011 [63], Sweden, case–control | Adult[b], population-based study (all ages), age/sex not reported, N = 7,969,645 | Suicide of relative (sibling, parent, or spouse) | Official records: Cause of death register. Exposure(s) determined by ICD codes for suicide and self-inflicted injury (ICD-8/9: E950–E959), intentional self-harm (ICD-10: X60–X84), and sequelae of intentional self-harm (ICD-10: Y870). | Suicide | Official death records: Cause of death register. Outcome(s) determined by ICD codes for suicide and self-inflicted injury (ICD-8/9: E950–E959), intentional self-harm (ICD-10: X60–X84), and sequelae of intentional self-harm (ICD-10: Y870). | The outcome occurred after the date of the exposure. | Good |
| Vijayakumar 1999 [69], India, case–control[a] | Adult[b], age range = 15 to 60+ (48.5% aged ≤24), 45.0% female, N = 200 | Suicide relative (any relative) | Informant: Informants were asked if there was a history of completed suicide in the family. Informed by family members. | Suicide | Official death records: Coroner-determined suicide death. | Psychological autopsy—suicide occurred after exposure. | Fair |
| Brent 2015 [49], US, cohort | Youth, mean age = 17.7, 48.1% female, N = 42 | Suicide attempt of relative (parent) | Informant: Informants were asked if a family member had made a suicide attempt, defined as a self-destructive act that resulted in potential or actual tissue damage with inferred or explicit intent to die. Informed by parents of cases and controls. | Suicide attempt | Self-report: Participants were asked if they had made a suicide attempt, defined as a self-destructive act that resulted in potential or actual tissue damage with inferred or explicit intent to die. | Number of new events of suicide attempt during 5-year follow-up period. | Good |

(*Continued*)

**Table 1.** (Continued)

| Study, location, study design | Exposed population, mean age or age range (years), percent female, total sample size | Exposure | Definition of exposure | Outcome(s) | Definition of outcome(s) | Exposure ascertainment | Study quality |
|---|---|---|---|---|---|---|---|
| Gould 1996 [54], US, case–control[a] | Youth, age range ≤ 18, 20.1% female, N = 267 | Suicide attempt of relative (parent) | Informant: Informants were asked if there was a history of first- and second-degree relatives who died by suicide or made a suicide attempt. Informed by parents or other adult who lived with the deceased. | Suicide | Official death records: Coroner-determined suicide death. | Psychological autopsy—suicide occurred after exposure. | Fair |
| Hu 2017 [48], Australia, case–control | Youth, age range = 10–19, 62.4% female, N = 150,171 | Suicide attempt of relative (parent) | Official records: Data linkage records for admission to hospital for deliberate self-harm. | Suicide attempt | Hospital admission records: Outcome(s) determined by ICD codes for suicide and self-inflicted injury (ICD-8/9: E950–E959), injury undetermined whether accidentally or purposely inflicted (ICD-8/9: E980–E989), intentional self-harm (ICD-10: X60–X84), and sequelae of intentional self-harm (ICD-10: Y870). | The outcome occurred after the date of the exposure determined through data linkage. | Good |
| Lewinsohn 1994 [56], US, cohort | Youth, mean age = 16.5, age range = 14–18, 54% female, N = 1,508 | Suicide attempt of friend or acquaintance | Self-report: Participants were asked if they knew a friend who had attempted suicide. | Suicide attempt | Self-report: Participants were asked if they have made an attempt to kill themselves. | Prior exposure measured at baseline, and subsequent suicide attempt was based on participants who reported having made a suicide attempt within the 1-year follow-up period. | Good |
| Mittendorfer-Rutz 2008 [62], Sweden, case–control | Youth, mean age = 19.1, 66.9% female, N = 158,840 | Suicide attempt of relative (first-degree relative) | Official records: Hospital admissions inpatient care register. Exposure(s) determined by ICD codes for suicide and self-inflicted injury (ICD-8/9: E950–E959), injury undetermined whether accidentally or purposely inflicted (ICD-8/9: E980–E989), intentional self-harm (ICD-10: X60–X84), and sequelae of intentional self-harm (ICD-10: Y870). | Suicide attempt | Hospital admission records: Outcome(s) determined by ICD codes for suicide and self-inflicted injury (ICD-8/9: E950–E959), injury undetermined whether accidentally or purposely inflicted (ICD-8/9: E980–E989), intentional self-harm (ICD-10: X60–X84), and sequelae of intentional self-harm (ICD-10: Y870). | All participants were hospitalized for deliberate self-harm at the time that prior exposure was measured. | Good |
| Nrugham 2008 [17], Norway, cohort | Youth, mean age = 14.9, age range = 15–20, 50.8% female, N = 265 | Suicide attempt of friend or acquaintance | Self-report: Participants were asked if they knew a friend who had attempted suicide. | Suicide attempt | Self-report: Participants were asked if they have ever tried to intentionally commit suicide. | Prior exposure measured at baseline, and subsequent suicide attempt was based on participants who reported having made a suicide attempt within the 1-year follow-up period. | Poor |
| Hishinuma 2018 [16], US, cohort | Youth, age range = 13–21, 54% female, N = 2,083 | Suicide attempt of relative (any relative) and suicide attempt of friend or acquaintance | Self-report: Participants were asked if a family member or friend had tried to commit suicide. | Suicide attempt | Self-report: Participants were asked if they had tried to commit suicide in the past 6 months (Major Life Events Scale). | Prior exposure measured at baseline, and subsequent suicide attempt was based on participants who reported having made a suicide attempt during the 5-year follow-up period. | Good |
| Ahmadi 2015 [39], Iran, case–control | Youth, mean age = 29 (60% aged ≤25), 76.0% female, N = 453 | Suicide of relative (first and second degree) and suicide attempt of relative (first and second degree) | Self-report: Suicide history in family and sibling, and parent's history of suicide attempt. | Suicide attempt | Hospital admission records: Hospital admission for deliberate self-inflicted immolation with suicide intent. | All participants were hospitalized for suicide attempt at the time that prior exposure was measured. | Fair |

(*Continued*)

**Table 1.** (*Continued*)

| Study, location, study design | Exposed population, mean age or age range (years), percent female, total sample size | Exposure | Definition of exposure | Outcome(s) | Definition of outcome(s) | Exposure ascertainment | Study quality |
|---|---|---|---|---|---|---|---|
| Chachamovich 2015 [52], Canada, case–control[a] | Youth, mean age = 23.4[d] age range = 1–25, 7.5% female, N = 240 | Suicide of relative (any relative); suicide attempt of relative (any relative) | Informant: Informants were asked if there was a history of suicide completion or suicide attempt in family. Informed by spouses, parents, or close friends of the deceased. | Suicide | Official death records: Coroner-determined suicide death. | Psychological autopsy—suicide occurred after exposure. | Fair |
| Chan 2018 [47], New Zealand, cross-sectional | Youth, age range = 13–19 (98.7% aged ≤17), 54.3% female, N = 8,497 | Suicide of relative (any relative) and friend or acquaintance; suicide attempt of relative (any relative) and friend or acquaintance | Self-report: Participants were asked if there was a history of suicide among their family or friends. For exposure to suicide attempt, participants were asked if anyone in their family or friends ever tried to kill themselves (attempted suicide?). | Suicidal ideation | Self-report: Participants were asked if they have made an attempt to kill themselves. | Exposure occurred >1 year ago, but ideation based on symptoms in the past year. | Fair |
| Garfinkel 1982 [53], Canada, case–control | Youth, mean age = 15.2, age range = 6–21, 75.4% female, N = 1,010 | Suicide of relative (parent); suicide attempt of relative (parent) | Official records: Chart review of family history of suicide attempts or suicide (completed suicide). | Suicide attempt | Hospital admission records: Hospital admission for suicide attempt with a conscious intent to die. | All participants were hospitalized for suicide attempt at the time that prior exposure was measured. | Poor |
| Palacio 2007 [70], Colombia, case–control[a] | Adult[b], median age = 29, 19.4% female, N = 216 | Suicide of relative (any relative); suicide attempt of relative (any relative) | Informant: Informants were asked if there was a history of suicide or suicide attempt in the family. Informed by relatives and medical documents. | Suicide | Official death records: Medical legal records of suicide cause of death. | Psychological autopsy—suicide occurred after exposure. | Poor |
| Thompson 2011 [18], US, cohort | Youth, mean age = 15.5, age range = 11–21, 49.1% female, N = 18,924 | Suicide of relative (any relative)[a] and friend or acquaintance; suicide attempt of relative (any relative)[a] and friend or acquaintance | Self-report: Participants were asked if a friend or family member had died by suicide. For exposure to suicide attempt, participants were asked if a friend or family member had made a suicide attempt. | Suicide attempt | Self-report: Participants were asked whether they had attempted suicide within the 12 months before the survey. | Prior exposure measured at baseline, and subsequent suicide attempt was based on participants who reported having made a suicide attempt during wave III (7 years later). | Fair |
| Phillips 2002 [68], China, case–control[a] | Adult[b], age range = 10 to 75+ (70% aged ≤30), 52% female, N = 1,055 | Suicidal behavior (composite) of relative (any relative) | Informant: Informants were asked if there was a family history of suicidal behavior (suicide attempts or suicide). Informed by family members of the deceased or close associates. | Suicide | Official death records: Medical legal records of suicide cause of death. | Psychological autopsy—suicide occurred after exposure. | Poor |
| Cheng 2000 [26], Taiwan, case–control[a] | Adult[b], mean age = 43.9, age range = 15–60, 39.8% female, N = 339 | Suicidal behavior (composite) of relative (any relative) | Informant: Informants were asked if there was a family history of suicidal behavior (suicide attempts or suicide). Informed by family members of the deceased. | Suicide | Official death records: Suicide as determined by prosecutor and coroner reports. | Psychological autopsy—suicide occurred after exposure. | Poor |
| Maniam 1994 [27], US, case–control[a] | Adult[b], mean age = 28.5, age range = 11–75, 50% female, N = 40 | Suicidal behavior (composite) of relative (any relative) | Informant: Informants were asked if there was a family history of suicidal behavior (suicide attempts or suicide). Informed by parents, spouses, or other adults who lived with the deceased. | Suicide | Official death records: Medical legal records of suicide cause of death. | Psychological autopsy—suicide occurred after exposure. | Poor |
| Jollant 2014 [65], US, case–control[a] | Youth, age range = 15–64 (56.25% aged ≤24), 25% female, N = 45 | Suicidal behavior (composite) of relative (any relative) | Informant: Informants were asked if there was a family history of suicidal behavior (suicide attempts or suicide). Informed by members of the community who knew the deceased. | Suicide | Informant: Suicide death reported by informants. | Psychological autopsy—suicide occurred after exposure. | Poor |

(*Continued*)

**Table 1.** (Continued)

| Study, location, study design | Exposed population, mean age or age range (years), percent female, total sample size | Exposure | Definition of exposure | Outcome (s) | Definition of outcome(s) | Exposure ascertainment | Study quality |
|---|---|---|---|---|---|---|---|
| Mercy 2001 [28], US, case–control | Youth, age range = 13–35 (50.3% aged ≤24), 54.5% female, $N$ = 666 | Suicidal behavior (composite) of relative (any relative); suicidal behavior (composite) of friend or acquaintance | Self-report: Participants were asked if their friends or family had committed suicide or made a suicide attempt. | Suicide attempt | Hospital admission records: Hospital admission for nearly lethal suicide attempt, defined as those in which the person probably would have died if they had not received emergency medical or surgical intervention or in which the attempter unequivocally used a method with a high case fatality ratio (i.e., a gun or a noose) and sustained an injury, regardless of severity. | All participants were hospitalized for suicide attempt at the time that prior exposure was measured. | Fair |

[a]Psychological autopsy study.

[b]Majority of the population aged >24 years and therefore categorized as adults.

[c]Exposure was a composite measure of suicide in a relative or friend; however, the majority were exposed to a friend's suicide.

[d]Exposure was a composite measure of suicidal behavior, but exposure to suicide was only 1%, and therefore the exposure was coded as exposure to suicide attempt.

ICD, International Classification of Diseases.

exposure to suicide and suicide attempt into a composite measure of exposure to suicidal behavior, did not provide adequate definitions of exposure to suicide or suicide attempt, and did not provide information on case ascertainment for suicide-related outcomes.

### Results of the multilevel meta-analysis

**Exposure to suicide.** Across 42 effect sizes from 22 studies, exposure to suicide was associated with 2.94-fold (95% CI = 2.30 to 3.75, $P <$ 0.001; Fig 2) increased odds of suicidal behavior (suicide or suicide attempt). Heterogeneity within and between studies was comparable ($\tau^2_{(2)}$ = 0.13, $I^2_{(2)}$ = 47%, 95% CI 15% to 94%; $\tau^2_{(3)}$ = 0.132, $I^2_{(3)}$ = 48%, 95% CI 1% to 81%). The funnel plot revealed evidence of asymmetry, which may indicate evidence of small study effect (Egger's intercept = 0.675, 1-tailed $P$ = 0.06; S1 Fig). Results from the subgroup analysis showed that exposure to suicide was associated with increased odds of suicide ($k$ = 24, OR = 3.23, 95% CI = 2.32 to 4.51, $P <$ 0.001) and suicide attempt ($k$ = 15, OR = 2.91, 95% CI = 2.01 to 4.23, $P <$ 0.001). However, there was no evidence of an association with suicidal ideation ($k$ = 3, OR = 1.85, 95% CI = 0.97 to 3.51, $P$ = 0.06; $Q$ between subgroups = 2.22, df = 2, $P$ = 0.33, $R^2_{(2)}$ = 11.8%, $R^2_{(3)}$ = 0%). The odds of later suicidal behavior were comparable when the exposure to suicide occurred in relatives ($k$ = 34, OR = 3.07, 95% CI = 2.35 to 4.01) and friends and acquaintances ($k$ = 8, OR = 2.42, 95% CI = 1.50 to 3.91; $Q$ = 0.77, df = 1, $P$ = 0.38, $R^2_{(2)}$ = 0%, $R^2_{(3)}$ = 2.7%). No further significant moderators relating to study design characteristics were identified (Table 2).

**Exposure to suicide attempt.** Across 19 effect sizes from 13 studies, exposure to suicide attempt was associated with 2.99-fold (95% CI = 2.19 to 4.09, $P < 0.001$; Fig 3) increased odds of suicidal behavior. Heterogeneity within studies was 9% ($\tau^2_{(2)} = 0.022$, $I^2_{(2)} = 9\%$, 95% CI 1% to 54%), while heterogeneity between studies was substantially larger ($\tau^2_{(3)} = 0.22$, $I^2_{(3)} = 88\%$, 95% CI 42% to 97%). Inspection of the funnel plot did not reveal evidence of small study effect (Egger's intercept = −0.453, $P = 0.33$; S2 Fig). Results from subgroup analysis revealed that exposure to suicide attempt was associated with greater odds of subsequent suicide attempt ($k = 16$, OR = 3.53, 95% CI = 2.63 to 4.73, $P < 0.001$) but not suicide death ($k = 3$, OR = 1.64, 95% CI = 0.90 to 2.98, $P = 0.10$; $Q$ between subgroups = 4.22, df = 1, $P = 0.04$, $R^2_{(2)} = 0\%$, $R^2_{(3)} = 3.8\%$). Significant between-group differences were observed for study design, with cross-sectional studies reporting greater odds of subsequent suicidal behavior ($k = 2$, OR = 8.23, 95% CI = 4.70 to 14.30, $P < 0.001$) compared to case–control studies ($k = 10$, OR = 2.74, 95% CI = 2.04 to 3.69, $P < 0.001$) and cohort studies ($k = 7$, OR = 2.69, 95% CI = 1.82 to 3.99, $P < 0.001$; $Q$ between subgroups = 7.35, df = 2, $P = 0.02$, $R^2_{(2)} = 0\%$, $R^2_{(3)} = 72.8\%$). Finally, moderator analyses revealed that psychological autopsy studies ($k = 3$, OR = 1.64, 95% CI = 0.90 to 2.99, $P = 0.127$) were associated with reduced odds of suicidal behavior compared to non-psychological autopsy studies ($k = 16$, OR = 3.53, 95% CI = 2.63 to 4.73, $P < 0.001$, $Q$-between subgroups = 4.22, df = 1, $P = 0.03$, $R^2_{(2)} = 0\%$, $R^2_{(3)} = 38.4\%$). No further significant differences were observed for the remaining moderators (Table 3).

**Exposure to suicidal behavior.** Across 10 effect sizes from 5 independent studies, exposure to suicidal behavior (composite measure—suicide or suicide attempt) was associated with 2.58-fold (95% CI = 1.25 to 5.35, $P = 0.01$) increased odds of suicidal behavior (Fig 4). Heterogeneity within and between studies was comparable ($\tau^2_{(2)} = 0.283$ $I^2_{(2)} = 38\%$; $\tau^2_{(3)} = 0.40$, $I^2_{(3)} = 53\%$). Visual inspection of the funnel plot did not reveal evidence of small study effect (S3 Fig). However, a formal test of asymmetry was not conducted due to insufficient studies. Results from the subgroup analysis revealed that exposure to suicidal behavior was associated with greater odds of suicide ($k = 7$, OR = 3.83, 95% CI = 2.38 to 6.17, $P < 0.001$) but not suicide attempt ($k = 3$, OR = 1.10, 95% CI = 0.69 to 1.76, $P = 0.90$; $Q$ between subgroups = 5.02, df = 1, $P = 0.02$, $R^2_{(2)} = 31.6\%$, $R^2_{(3)} = 100\%$). The odds of suicidal behavior were also greater when the exposure occurred in relatives ($k = 8$, OR = 3.09, 95% CI = 1.53 to 6.26, $P = 0.001$) compared to friends and acquaintances ($k = 2$, OR = 1.33, 95% CI = 0.69 to 2.92, $P = 0.48$; $Q$ between subgroups = 5.20, df = 1, $P = 0.02$, $R^2_{(2)} = 86.2\%$, $R^2_{(3)} = 0\%$). No significant differences were observed for the remaining moderators (Table 4).

## Discussion

Based on findings from 34 studies of mostly good and fair quality, encompassing 13,923,029 participants and 71 effect sizes, we found that prior exposure to suicide was associated with significantly greater odds of suicidal behavior (suicide or suicide attempt; OR = 2.94). Results of the moderator analysis revealed that prior exposure to suicide was associated with 3.23-fold increased odds of suicide and 2.91-fold increased odds of suicide attempt, while there was no evidence of an association between exposure to suicide and subsequent suicidal ideation. These findings remained robust across cohort, case–control, and cross-sectional studies, as well as exposure and outcome measurements encompassing informant interview, self-report, and official records (e.g., coroner reports, hospital admission records, or data linkage with birth and death registries).

Exposure to suicide attempt was associated with increased odds of suicidal behavior (OR = 2.99). However, moderator analyses revealed that the association of exposure to suicide

attempt with suicide-related outcomes was significant only for suicide attempt (OR = 3.53), not for suicide death (OR = 1.64). These findings were demonstrated across 19 effect sizes from 13 studies of mostly fair quality, and corroborated by 3 large population-based studies using data linkage or hospital admission records for suicide attempt [48,59,62]. Exposure to suicidal behavior (suicide or suicide attempt) was associated with a 2.58-fold increased odds of suicidal behavior, but moderator analysis revealed that this was significant only for outcomes relating to suicide death (OR = 3.83), not suicide attempt (OR = 1.10). These findings were demonstrated across 10 effect sizes from 5 studies, the majority of which involved psychological autopsy methodologies.

Our analyses update and further specify the findings from previous systematic reviews, which included estimates from studies reporting lifetime prevalence or did not differentiate between the independent effects associated with exposure to suicide and exposure to suicide attempt [19,22,23]. The finding that exposure to suicide was associated with an increased odds of suicide and suicide attempt—in contrast to exposure to suicide attempt, which was associated with an increased odds of suicide attempt only—indicates that exposure to suicide and suicide attempt do not incur uniform risk across the range of suicide-related outcomes. This was corroborated by our analysis of exposure to suicidal behavior, which found that this composite measure was associated with increased odds of suicide but not suicide attempt, a finding that was inconsistent with our separate analyses of exposure to suicide and exposure to suicide attempt. Taken together, the present findings raise questions about the conceptual value of combining suicide and suicide attempt as a composite measure of suicidal behavior, and suggest that future research and public health policies should refrain from combining these exposures and outcomes into 1 composite measure of suicidal behavior.

Evidence from 2 studies [46,57] suggests that exposure to suicide may be associated with increased risk of suicidal ideation, especially in older adults [46]. Conversely, results from a single cohort study in youths [57] indicate higher risk for suicide attempt than for suicidal ideation, pointing once more to lack of uniformity across populations and outcomes. Moreover, theoretical and empirical accounts suggest that while exposure to suicide may contribute to subsequent suicidal ideation to some extent, its effect on people with a history of suicidal ideation may be more pronounced [71], as this experience might reduce cognitive and practical barriers to acting on one's suicidal thoughts [46,72,73]. A more comprehensive look at this interaction may have important practical implications for developing specific interventions for this high-risk population, in particular interventions guided by the "ideation-to-action framework" [71] that aim to reduce acquired capability for suicidal behavior among individuals exposed to suicide.

The increased risks associated with exposure to suicide for outcomes relating to suicide and suicide attempt in the current meta-analyses suggest that further consideration should be given towards developing interventions that target suicide-related outcomes in those bereaved by suicide. To date, interventions targeting those exposed to suicide have largely focused on bereavement-related factors such as grief, reduced social support, and stigma [74,75]. Although previous studies have shown that these factors are elevated among those bereaved by suicide as opposed to other modes of death, there remains a dearth of studies that investigate the effectiveness of interventions on suicide and suicide attempt behavior. A recent review by Andriessen and colleagues [74], for example, found 3 controlled studies [76–78] that investigated the effectiveness of an intervention on suicidal ideation and found no studies that included outcomes related to suicide or suicide attempt.

Although we did not observe a significant association between exposure to suicide attempt and subsequent suicide, the specific relationship between exposure to suicide attempt and subsequent suicide attempt is noteworthy, since suicide attempt is associated with significant

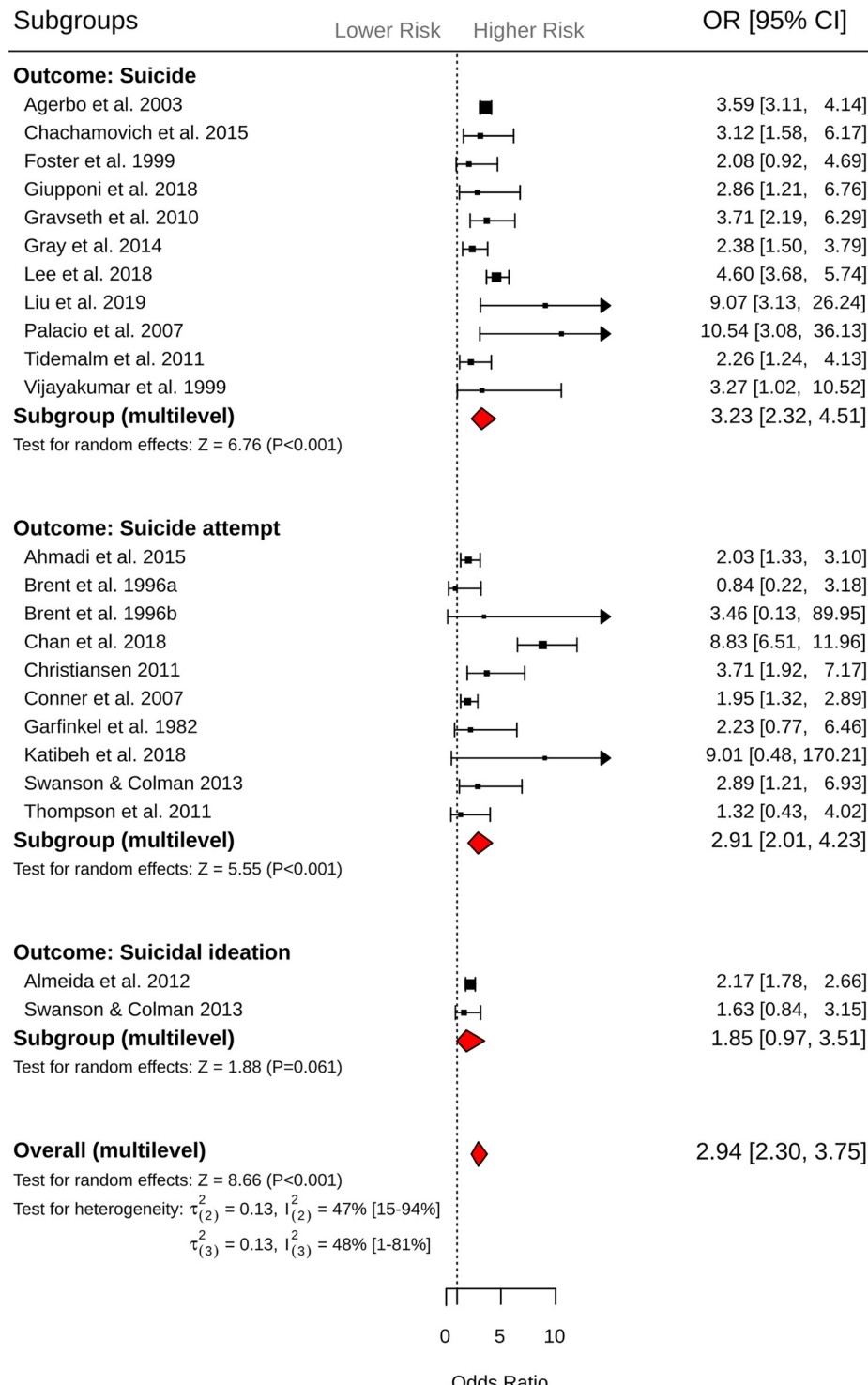

**Fig 2. Forest plots of exposure to suicide and subsequent suicide, suicide attempt, and suicidal ideation outcomes.**
CI, confidence interval; OR, odds ratio.

disruptions to an individual's milieu, and has been linked to adverse psychosocial and mental health stressors that persist later in life [5]. The findings from our analysis of exposure to

**Table 2. Results of moderator analyses of exposure to suicide across suicide, suicide attempt, and suicidal ideation outcomes.**

| Moderator | Number of effect sizes | Odds ratio (95% confidence interval) | P value | $R^2_{(2)}$ | $R^2_{(3)}$ | ANOVA between-group P value |
|---|---|---|---|---|---|---|
| Proximity | | | | | | |
| Relative | 34 | 3.07 (2.35 to 4.01) | <0.001 | | | |
| Friend or acquaintance | 8 | 2.42 (1.50 to 3.91) | <0.001 | <0.001 | 0.03 | 0.38 |
| Population at risk | | | | | | |
| Adult | 24 | 2.80 (2.00 to 3.92) | <0.001 | | | |
| Youth | 18 | 3.04 (2.14 to 4.32) | <0.001 | <0.001 | 0.06 | 0.74 |
| Outcome measurement | | | | | | |
| Informant interview | 2 | 1.53 (0.63 to 3.73) | 0.35 | | | |
| Official records | 30 | 3.10 (2.30 to 4.17) | <0.001 | | | |
| Self-report | 10 | 2.97 (1.86 to 4.75) | <0.001 | 0.04 | 0.04 | 0.34 |
| Exposure measurement | | | | | | |
| Informant interview | 7 | 3.53 (2.13 to 5.83) | <0.001 | | | |
| Official records | 20 | 2.84 (1.93 to 4.18) | <0.001 | | | |
| Self-report | 15 | 2.66 (1.78 to 3.97) | <0.001 | <0.01 | <0.01 | 0.68 |
| Psychological autopsy | | | | | | |
| No | 34 | 2.64 (2.64 to 3.50) | <0.001 | | | |
| Yes | 8 | 3.71 (2.38 to 5.78) | <0.001 | 0.03 | 0.07 | 0.21 |
| Study design | | | | | | |
| Case–control | 29 | 2.85 (2.14 to 3.80) | <0.001 | | | |
| Cohort | 10 | 2.13 (1.35 to 3.36) | 0.01 | | | |
| Cross-sectional | 3 | 4.98 (2.73 to 9.08) | <0.001 | <0.01 | 0.47 | 0.12 |
| Study quality | | | | | | |
| Good | 23 | 2.61 (1.86 to 3.67) | <0.001 | | | |
| Fair | 15 | 3.03 (2.13 to 4.30) | <0.001 | | | |
| Poor | 4 | 5.15 (1.97 to 13.48) | <0.001 | <0.001 | 0.09 | 0.41 |

ANOVA, analysis of variance.

suicide attempt also provide some insight into the mechanisms underlying the observed association between exposure to suicide and exposure to suicide attempt and the suicide-related outcomes. Arguably, the absence of bereavement-related factors and the specific association between exposure to suicide attempt and subsequent suicide attempt support the hypothesis that suicidal individuals may model, or imitate, suicide-related behavior that they see in others [10]. An imitation model is consistent with previous studies that have shown that increased risk of suicide-related behavior following exposure to both suicide and suicide attempt is not significantly moderated by preexisting risk factors such as depression, anxiety, and hospital admission for mental health [79,80]. The finding that exposure to suicide is associated with significant increased odds of suicide attempt is important since public health approaches for the prevention of behavioral contagion of both suicide and suicide attempt, such as frameworks for the prevention of suicide and self-harm clusters [12–15], have focused largely on mitigation efforts following exposure to suicide and therefore may benefit from the inclusion of exposure to suicide attempt in future mitigation efforts.

## Limitations

The current systematic review and meta-analysis is the first to our knowledge to quantify the association between exposure to suicide and suicide attempt and the full spectrum of suicide-

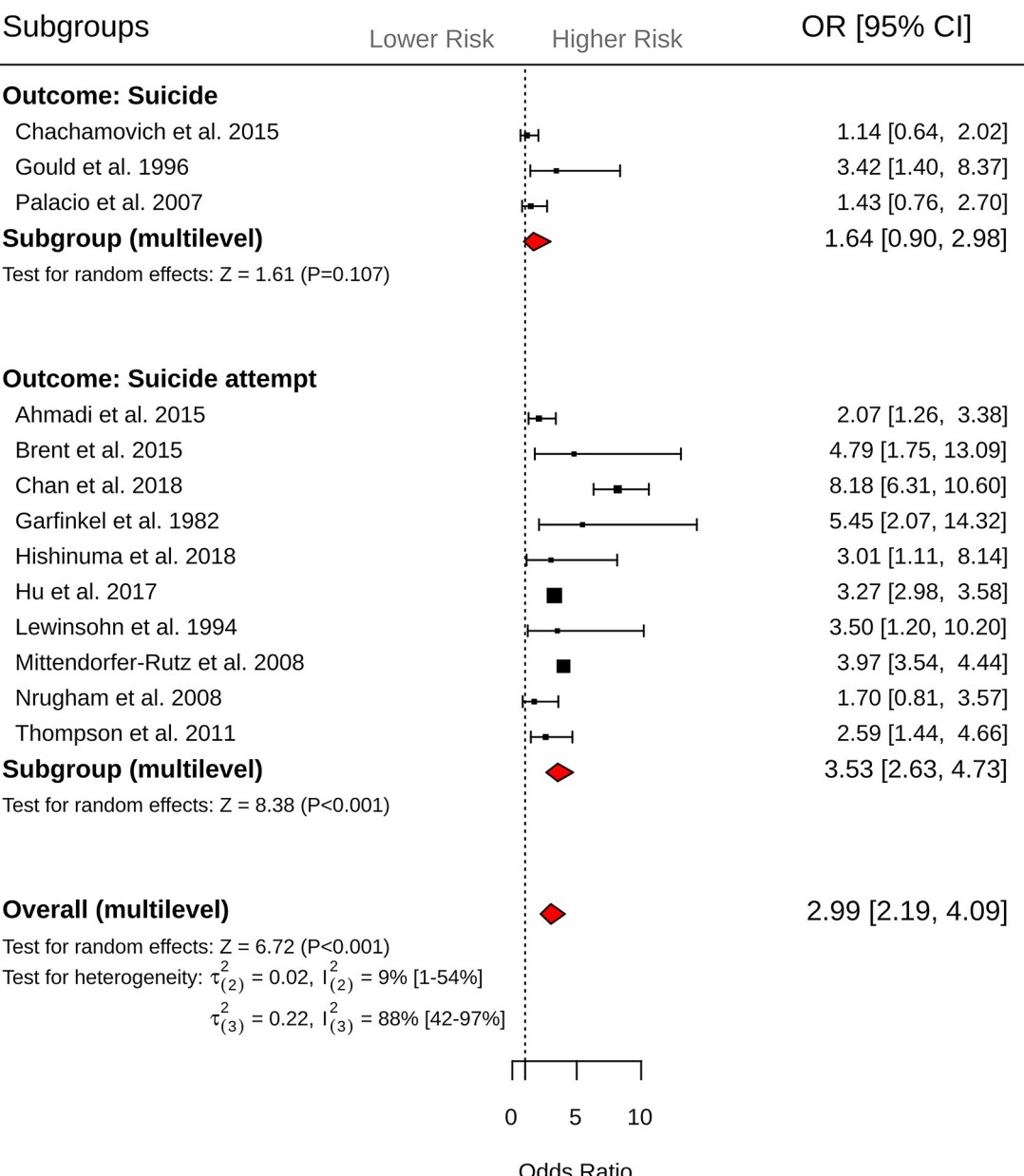

**Fig 3. Forest plots of exposure to suicide attempt and subsequent suicide and suicide attempt outcomes.** CI, confidence interval; OR, odds ratio.

related outcomes and has many strengths, including the use of multilevel meta-analysis, the large sample size, and the exclusion of estimates of lifetime prevalence that do not take into account the temporal sequence between exposure and suicide-related outcomes. Despite this, several limitations exist. Whilst we conducted an extensive search of 21,868 records, there is the possibility that some relevant studies were not detected. Such studies are likely to create a bias towards the null (i.e., the exposure not having a significant effect). This is a limitation that is common to many systematic reviews and was mitigated to the best of our ability through adherence to a screening protocol developed a priori.

**Table 3. Results of moderator analyses of exposure to suicide attempt across suicide, suicide attempt, and suicidal ideation outcomes.**

| Moderator | Number of effect sizes | Odds ratio (95% confidence interval) | $P$ value | $R^2_{(2)}$ | $R^2_{(3)}$ | ANOVA between-group $P$ value |
|---|---|---|---|---|---|---|
| Proximity | | | | | | |
| Relative | 14 | 3.14 (2.25 to 4.38) | <0.001 | | | |
| Friend or acquaintance | 5 | 2.64 (1.72 to 4.03) | <0.001 | 0.14 | <0.001 | 0.39 |
| Population at risk | | | | | | |
| Adult | 1 | 1.43 (0.48 to 4.32) | 0.52 | | | |
| Youth | 18 | 3.19 (2.35 to 4.32) | <0.001 | <0.001 | 0.16 | 0.18 |
| Outcome measurement | | | | | | |
| Official records | 10 | 2.60 (1.75 to 3.87) | <0.001 | | | |
| Self-report | 9 | 3.62 (2.30 to 5.68) | <0.001 | <0.01 | 0.13 | 0.29 |
| Exposure measurement | | | | | | |
| Informant interview | 3 | 1.64 (0.90 to 2.99) | 0.12 | | | |
| Official records | 5 | 3.60 (2.12 to 6.10) | <0.001 | | | |
| Self-report | 11 | 3.49 (2.45 to 4.98) | <0.001 | 0.01 | 0.38 | 0.12 |
| Psychological autopsy | | | | | | |
| No | 16 | 3.53 (2.63 to 4.73) | <0.001 | | | |
| Yes | 3 | 1.64 (0.90 to 2.99) | 0.13 | <0.001 | 0.38 | 0.03 |
| Study design | | | | | | |
| Case–control | 10 | 2.74 (2.04 to 3.69) | <0.001 | | | |
| Cohort | 7 | 2.69 (1.82 to 3.99) | <0.001 | | | |
| Cross-sectional | 2 | 8.23 (4.70 to 14.30) | <0.001 | 0.01 | 0.73 | 0.02 |
| Study quality | | | | | | |
| Good | 7 | 3.74 (2.20 to 6.30) | <0.001 | | | |
| Fair | 9 | 2.95 (1.93 to 4.50) | <0.001 | | | |
| Poor | 3 | 2.18 (1.10 to 4.32) | 0.02 | 0.02 | 0.18 | 0.48 |

ANOVA, analysis of variance.

Furthermore, since most studies adjusted for different covariates, we restricted our analysis to unadjusted events and ORs. Whilst this is consistent with previous meta-analyses in the field [81,82], it meant that we could not investigate other risk factors, such as frequency of exposure, duration since exposure, and baseline mental health diagnoses, and how these might moderate the association between exposure to suicide and suicide attempt and suicide-related outcomes. For example, a previous systematic review on pre- and post-loss features of suicide bereavement in young people found evidence of a cumulative effect of exposure to suicide on subsequent suicide risk [83]. In the present meta-analysis, 2 out of 34 studies included in our analyses provided separate estimates for multiple exposures to suicide [62] and suicide attempt [48]. In 1 study [62], exposure to 2 or more suicide deaths affected less than 1% of the population, but was associated with 9.8-fold greater odds of suicide attempt, compared to an OR of 3.8 among those who had been exposed to the suicide of 1 relative. Similarly, those exposed to the suicide attempt of 2 parents were 5.67 times more likely to make a suicide attempt, compared to ORs of 2.89 and 3.89 (for paternal and maternal exposures, respectively) among youths who had been exposed to the suicide attempt of 1 parent [48].

Indeed, in the present multilevel meta-analysis, within-study heterogeneity remained largely unchanged by study-level moderators for both exposure to suicide and exposure to suicide attempt. For example, we did not find evidence of a significant difference in suicide-related outcomes when the exposure to suicide or suicide attempt occurred in relatives compared to friends and acquaintances. Although previous registry-based studies have shown a

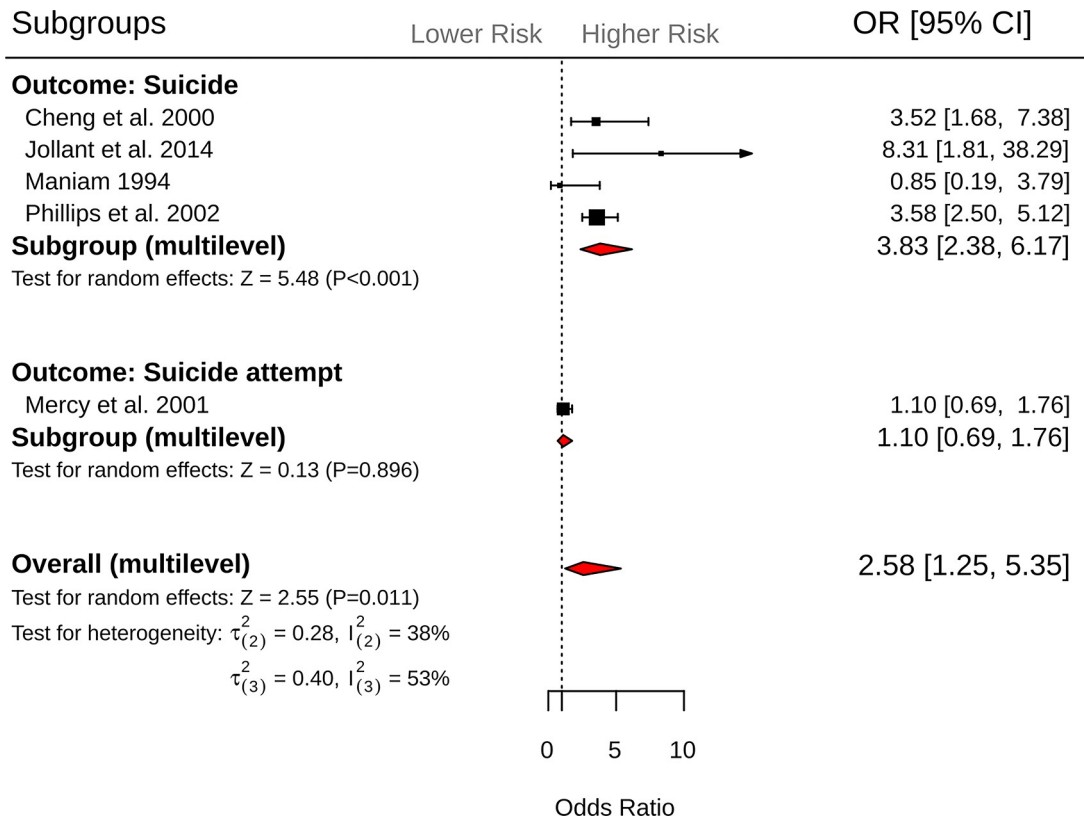

**Fig 4. Forest plots of exposure to suicidal behavior (composite measure—suicide or suicide attempt) and subsequent suicide and suicide attempt outcomes.** CI, confidence interval; OR, odds ratio.

6-fold increase of suicide among biological relatives of adoptees who have died by suicide [84], in the present meta-analysis it was not possible to delineate between relatives who resided in the same household, and therefore shared many of the same environmental risk factors, and relatives who did not [9]. Understanding these factors is important for identifying specifically who within in the general population is most at risk. However, the pooling of observational studies meant that analyses of these factors were outside the scope of the present study. An important next step forward would therefore be examinations of exposure to suicide and suicide attempt while taking these risk factors into account using individual participant data meta-analyses.

In the present multilevel meta-analysis, between-study heterogeneity remained moderate ($I^2_{(3)}$ = 52.2%) across studies measuring exposure to suicide, which was not sufficiently explained by any of the included study design moderators. By contrast, study design characteristics accounted for 72.8% of between-study heterogeneity ($I^2_{(3)}$ = 87.8%) across studies measuring exposure to suicide attempt. In this instance, cross-sectional studies reported significantly larger ORs (OR = 8.23) compared to case–control (OR = 2.74) and cohort (OR = 2.69) studies. In general, cross-sectional studies are prone to an inherently greater number of biases, compared to case–control and cohort studies. This may be particularly pronounced in studies that measure suicide attempt because recall of suicide attempt may be less salient than recall of suicide death, and is prone to multiple interpretations and definitions [85].

It is noteworthy that we did not find evidence to support the role of age as a risk moderator, as suggested in previous reviews [9,10]. Yet these results should be interpreted with caution, as

**Table 4. Results of moderator analyses of exposure to suicidal behavior (composite measure—suicide or suicide attempt) across suicide, suicide attempt, and suicidal ideation outcomes.**

| Moderator | Number of effect sizes | Odds ratio (95% confidence interval) | P value | $R^2_{(2)}$ | $R^2_{(3)}$ | ANOVA between-group P value |
|---|---|---|---|---|---|---|
| Proximity | | | | | | |
| Relative | 8 | 3.09 (1.53 to 6.26) | 0.001 | | | |
| Friend or acquaintance | 2 | 1.33 (0.60 to 2.92) | 0.48 | 0.86 | <0.001 | 0.02 |
| Population at risk | | | | | | |
| Adult | 4 | 2.63 (0.96 to 7.22) | 0.06 | | | |
| Youth | 6 | 2.53 (0.86 to 7.43) | 0.09 | <0.001 | 0.01 | 0.96 |
| Outcome measurement | | | | | | |
| Informant interview | 3 | 8.34 (2.35 to 29.63) | 0.01 | | | |
| Official records | 7 | 1.90 (1.03 to 3.50) | 0.05 | <0.01 | 0.70 | 0.07 |
| Exposure measurement | | | | | | |
| Informant interview | 7 | 3.83 (2.37 to 6.19) | <0.001 | | | |
| Self-report | 3 | 1.04 (0.59 to 1.83) | 0.89 | 0.32 | 1 | 0.02 |
| Psychological autopsy | | | | | | |
| No | 3 | 1.04 (0.59 to 1.83) | 0.89 | | | |
| Yes | 7 | 3.83 (2.37 to 6.19) | <0.001 | 0.32 | 1 | 0.02 |
| Study quality | | | | | | |
| Fair | 3 | 1.04 (0.59 to 1.83) | 0.89 | | | |
| Poor | 7 | 3.83 (2.37 to 6.19) | <0.001 | 0.32 | 1 | 0.02 |

ANOVA, analysis of variance.

the dichotomization of study populations into the categories of youths and adults was based on a majority rule in 13 out of 34 studies [26–28,39,58,60,61,63–65,68–70]. The finding that age was not a risk moderator may therefore be an artifact introduced by the imprecise age classification of the included population in individual studies. Furthermore, whilst similar patterns were observed across studies examining exposure to suicide attempt in youths versus adults, only 1 out of 13 studies [70] reported outcomes among adults, which may have impacted our ability to detect a statistically significant difference.

Finally, the results of the present study do not allow causality to be inferred, and although we show evidence of a temporal association between prior exposure to suicide and suicide attempt and subsequent suicide-related outcomes, cross-sectional studies, by virtue of study design, do not provide incidence estimates. To account for this limitation, we only included cross-sectional studies where participants were explicitly asked about suicidal acts that occurred after exposure to suicide or suicide attempt. But this approach does not mitigate errors in recall and other biases that are inherently more common in cross-sectional studies.

## Conclusions

Our findings suggest that prior exposure to suicide is associated with increased risk of suicide and suicide attempt. By contrast, exposure to suicide attempt is associated with increased risk of suicide attempt, but not suicide death. Future studies should refrain from combining suicidal behaviors into composite measures of suicide exposures and outcomes as the relationships between exposure to suicide and suicide attempt and suicide-related outcomes are markedly different. Lastly, future studies should consider interventions that target suicide-related outcomes in those exposed to suicide and include efforts to mitigate the adverse effects associated with exposure to suicide attempt.

## Supporting information

**S1 Data. Summary data for all included studies.**
(XLS)

**S1 Fig. Exposure to suicide funnel plot.** The solid vertical lines indicate the 95% confidence interval around the log odds ratio (LogOR). The dashed lines indicate the summary log odds ratio ± 1.96 × standard error for each of the standard errors on the $y$-axis. The resulting triangular region indicates the expected location of 95% of studies in the absence of small study effect.
(TIF)

**S2 Fig. Exposure to suicide attempt funnel plot.** The solid vertical lines indicate the 95% confidence interval around the log odds ratio (LogOR). The dashed lines indicate the summary log odds ratio ± 1.96 × standard error for each of the standard errors on the $y$-axis. The resulting triangular region indicates the expected location of 95% of studies in the absence of small study effect.
(TIFF)

**S3 Fig. Exposure to suicidal behavior funnel plot.** The solid vertical lines indicate the 95% confidence interval around the log odds ratio (LogOR). The dashed lines indicate the summary log odds ratio ± 1.96 × standard error for each of the standard errors on the $y$-axis. The resulting triangular region indicates the expected location of 95% of studies in the absence of small study effect.
(TIFF)

**S1 Table. Description of a priori study moderators used for data extraction.**
(DOCX)

**S2 Table. Excluded overlapping studies.**
(DOCX)

**S3 Table. Risk of bias.**
(DOCX)

**S1 Text. PRISMA checklist.**
(DOC)

**S2 Text. MEDLINE search strategy.**
(DOCX)

**S3 Text. Articles excluded from the systematic review and meta-analysis.**
(DOCX)

## Author Contributions

**Conceptualization:** Nicole T. M. Hill, Jo Robinson, Allison Milner, Katrina Witt, Amit Lampit.

**Data curation:** Nicole T. M. Hill, Karl Andriessen, Karolina Krysinska, Amber Payne, Alexandra Boland, Alison Clarke, Katrina Witt, Stephan Krohn.

**Formal analysis:** Nicole T. M. Hill, Stephan Krohn, Amit Lampit.

**Investigation:** Nicole T. M. Hill, Karolina Krysinska, Stephan Krohn, Amit Lampit.

**Methodology:** Nicole T. M. Hill, Jane Pirkis, Karl Andriessen, Karolina Krysinska, Allison Milner, Katrina Witt, Stephan Krohn, Amit Lampit.

**Project administration:** Nicole T. M. Hill.

**Resources:** Nicole T. M. Hill.

**Software:** Nicole T. M. Hill, Stephan Krohn, Amit Lampit.

**Supervision:** Nicole T. M. Hill, Jo Robinson, Jane Pirkis, Allison Milner, Amit Lampit.

**Validation:** Nicole T. M. Hill, Karl Andriessen, Karolina Krysinska, Amber Payne, Alexandra Boland, Stephan Krohn, Amit Lampit.

**Visualization:** Nicole T. M. Hill, Stephan Krohn, Amit Lampit.

**Writing – original draft:** Nicole T. M. Hill, Jo Robinson, Jane Pirkis, Karl Andriessen, Karolina Krysinska, Amber Payne, Alexandra Boland, Alison Clarke, Allison Milner, Katrina Witt, Stephan Krohn, Amit Lampit.

**Writing – review & editing:** Nicole T. M. Hill, Jo Robinson, Jane Pirkis, Karl Andriessen, Karolina Krysinska, Amber Payne, Alexandra Boland, Alison Clarke, Allison Milner, Katrina Witt, Stephan Krohn, Amit Lampit.

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
