## [Decision Letter · Decision Letter 0]

19 Nov 2019

Dear Dr. Lampit,

Thank you very much for submitting your manuscript "Risk of suicidal behavior following exposure to suicide and suicide attempt: A systematic review and multilevel meta-analysis" (PMEDICINE-D-19-02935) for consideration at PLOS Medicine. 

Your paper was evaluated by a senior editor and discussed among all the editors here. It was also discussed with an academic editor with relevant expertise, and sent to three independent reviewers, including a statistical reviewer. The reviews are appended at the bottom of this email and the accompanying attachment from reviewer 2 can be seen via the link below:

[LINK]

In light of these reviews, I am afraid that we will not be able to accept the manuscript for publication in the journal in its current form, but we would like to consider a revised version that addresses the reviewers' and editors' comments. Obviously we cannot make any decision about publication until we have seen the revised manuscript and your response, and we plan to seek re-review by one or more of the reviewers. 

We expect to receive your revised manuscript by Dec 10 2019 11:59PM. Please email us (plosmedicine@plos.org) if you have any questions or concerns.

We look forward to receiving your revised manuscript. 

Sincerely,

Caitlin Moyer, Ph.D.

Associate Editor 

PLOS Medicine

plosmedicine.org

1. Data availability statement: Thank you for your willingness to make your data available as Supporting Information. Please note in the Data Availability Statement and the Methods that the study data are included as Supporting Information, and reference the file (e.g. S1 Table).

2. Title: Please revise the title to: “Association of suicidal behavior with exposure to suicide and suicide attempt: A systematic review and multilevel meta-analysis

3. Abstract: Please include methods of assessing risk of bias in the abstract Methods and Findings section.

4. Abstract: Please include numbers of studies and participants for each of the main outcomes described in the abstract Methods and Findings section.

5. Abstract: Please quantify the main results with both 95% CIs and p values.

6. Author Summary: At this stage, we ask that you include a short, non-technical Author Summary of your research to make findings accessible to a wide audience that includes both scientists and non-scientists. 

The Author Summary should immediately follow the Abstract in your revised manuscript. This text is subject to editorial change and should be distinct from the scientific abstract. Please see our author guidelines for more information: https://journals.plos.org/plosmedicine/s/revising-your-manuscript#loc-author-summary

7. Methods: Please update your search to the present time, as we require that SRs are updated to within roughly 6 months of the expected publication date and your search has not been updated since June 2019.

8. Results: Please provide both 95% CIs and p values for the analysis of exposure to suicide and association with increased suicide, suicide attempt, and suicide ideation (Lines 275-277).

9. Discussion: Conclusions: Line 452-425: Please revise the first sentence of the conclusion paragraph to: “Our findings suggest that prior exposure to suicide is associated with increased risk of suicide and suicide attempt.” or similar.

10. Figure 2, Figure 3, Figure 4: Please provide p values associated with the 95% CIs and odds ratios presented. Please define the abbreviation “OR” and “CI” in the figure legend. Please indicate in the figure legend the meaning of the bars and markers.

11.Table 2, Table 3: Please define the abbreviation “CI” in the table legend.

12. S6 Figure and S7 Figure: One of the figures appears to have been duplicated here, please update with the correct figures.

13. S6, S7, and S8 Figures: Please define the dashed and solid lines in the funnel plots in the figure legends. Please define the abbreviation “OR” in the figure legends.

14. S9 Figure: What is the rationale for including the results of moderator analyses for exposure to suicide and suicide attempt as main manuscript tables, while the moderator analyses for exposure to composite measures of suicidal behavior is included in the supporting information? Please include these data in the main manuscript unless there is a reason not to do so.

15. Checklist: Thank you for completing and including the PRISMA Checklist (S10 Text). Please revise the checklist using section and paragraph numbers, rather than page numbers, to note the location of the checklist items.

Comments from the reviewers:

Reviewer #1: This article describes the results of a three level meta-analysis aimed at assessing the risk associated with exposure to suicide and suicide attempt on suicide, suicide attempt and suicidal ideation and to identify moderators of this risk. The authors found that prior exposure to suicide and suicide attempt is associated with increased risk of suicidal behaviour. The topic is worthwhile: form a public health perspective it is important to investigate risk factors of suicide. Such examination will lead to the development of targeted preventive interventions to reduce the risk of suicide. 

Although this study has been conducted thoroughly, my main concern lies on pooling together rates from observation studies. It is well known that observational studies are susceptible to selection bias and their results are most of the times influenced by several confounding variables. The authors tried to adjust for confounders but there might be many more factors that have not been taking into account when pooling the rates across the different studies. Moreover, the gold standard approach in examining moderators at a meta-analytic level is the individual patient data meta-analysis. Study level meta-analysis rely on average effects and cannot examine individual participant differences. Thus, the examination of moderators is very limited in a conventional meta-analysis. Moreover, I have some additional comments related to heterogeneity: 

(a) in many outcomes, heterogeneity is surprisingly "low" given the nature of the included studies. Nevertheless, one cannot fully grasp the magnitude of heterogeneity based solely on the point estimate of I-square. The authors should calculate and report 95% CI around I-square to allow a better understanding of the heterogeneity in their meta-analyses.

(b) the authors reported a very high heterogeneity (87%) across 13 studies examining exposure to suicide attempt. Such heterogeneity implies a great diversity among the examined studies, making the use of meta-analytic methodology questionable. Did the authors explore sources of this heterogeneity? How do they interpret it? 

Reviewer #2: See attachment

Michael Dewey

Reviewer #3: This brief study provides very useful information about exposure to suicide and suicide attempts. I believe it represents a meaningful contribution to the literature. My brief comments have to do with enhancing interpretation of results.

First, it is noteworthy that exposure to suicide/suicide attempt in relatives vs. non-relatives produce similar odds ratios. This pattern suggests that the mechanism of increased risk for suicidal behavior is likely environmental more than genetic. That said, there was a tendency for the relatives odds ratio to be modestly higher than for non-relatives; if the authors think this difference is meaningful (it appears to be on the margins of statistical reliability), then perhaps the data suggest a small role for genetic transmission of risk in addition to environment.

Second, it is noteworthy that exposure increases risk for suicidal behavior but not attempts. This pattern is consistent with recent suicide theory emphasizing a role for suicide capability in the transition from suicidal thoughts and behavior. For example the Three Step Theory (Klonsky & May, 2015) suggests that pain and hopelessness drive suicidal ideation, but capability for suicide (including acquired and practical contributors to capability) increase one's ability to act on suicidal ideation and make an actual suicide attempt. From this perspective, it is significant that exposure increases risk for behavior rather than ideation. Most people with suicidal thoughts do not make an attempt. For most people, even people with suicidal ideation, the idea of attempting suicide is cognitive difficult (is this really something i could do?) as well as practically difficult (how would i do it? what if my attempt went wrong? what if it's too scary?). Being exposed to someone who has died by suicide can make the act of suicide seem more plausible, more cognitively feasible. In addition, knowing the method of the person's attempt can increase practical capability, in that the person exposed to the attempt now has concrete information about a method that resulted in suicide death. Thus, from the perspective of the Three Step Theory, exposure to suicide increases capability for suicide which, in turn, increases risk of suicidal behavior (but not suicide ideation).

[LINK]

---

## [Decision Letter · Decision Letter 1]

3 Feb 2020

Dear Dr. Lampit,

Thank you very much for re-submitting your manuscript "Association of suicidal behavior with exposure to suicide and suicide attempt: A systematic review and multilevel meta-analysis" (PMEDICINE-D-19-02935R1) for review by PLOS Medicine.

I have discussed the paper with my colleagues and the academic editor and it was also seen again by one of the reviewers. I am pleased to say that provided the remaining editorial and production issues are dealt with we are planning to accept the paper for publication in the journal.

[LINK]

We look forward to receiving the revised manuscript by Feb 10 2020 11:59PM. 

Sincerely,

Caitlin Moyer, Ph.D.

Associate Editor 

PLOS Medicine

plosmedicine.org

Requests from Editors:

The abstract could be trimmed – perhaps the methods and findings section, mostly.

 Thank you for providing an author summary – it should be in bulleted format for house style, please (the text of the points is fine, just please add bullets)

Line 293 Studies were most commonly rated fair (13/34), followed by good (13/34) and poor – is this correct? Fair and good have the same rating? Just checking, apologies if I’ve misunderstood.

Line 505 – please just say association instead of evidence of association

Line 36 – limitations – Please note study quality, and/or the possibility of unmeasured confounders influencing the findings

 At line 40, suggest beginning the sentence with "The findings of this systematic review and meta-analysis indicate that ..."

Please delete "2.99" at line 377 (or add "(OR=2.99)"). 

I wonder if "prior exposure to suicide" is somewhat ambiguous as it could be read to include exposure to media representations. Perhaps one might amend the wording to "direct exposure to suicide" early on in the paper.

Comments from Reviewers:

Reviewer #2: The authors have addressed all my points.

Michael Dewey

[LINK]

---

## [Editor Report · Decision Letter 2]

21 Feb 2020

Dear Dr Lampit, 

On behalf of my colleagues and the academic editor, Dr. Vikram Patel, I am delighted to inform you that your manuscript entitled "Association of suicidal behavior with exposure to suicide and suicide attempt: A systematic review and multilevel meta-analysis" (PMEDICINE-D-19-02935R2) has been accepted for publication in PLOS Medicine. 

PRODUCTION PROCESS

PRESS

PROFILE INFORMATION

Thank you again for submitting the manuscript to PLOS Medicine. We look forward to publishing it. 

Best wishes, 

Caitlin Moyer, Ph.D.

Associate Editor 

PLOS Medicine

plosmedicine.org